# Photonic Weyl point in a two-dimensional resonator lattice with a synthetic frequency dimension

Qian Lin[1], Meng Xiao[2], Luqi Yuan[2] & Shanhui Fan[2]

Weyl points, as a signature of 3D topological states, have been extensively studied in condensed matter systems. Recently, the physics of Weyl points has also been explored in electromagnetic structures such as photonic crystals and metamaterials. These structures typically have complex three-dimensional geometries, which limits the potential for exploring Weyl point physics in on-chip integrated systems. Here we show that Weyl point physics emerges in a system of two-dimensional arrays of resonators undergoing dynamic modulation of refractive index. In addition, the phase of modulation can be controlled to explore Weyl points under different symmetries. Furthermore, unlike static structures, in this system the non-trivial topology of the Weyl point manifests in terms of surface state arcs in the synthetic space that exhibit one-way frequency conversion. Our system therefore provides a versatile platform to explore and exploit Weyl point physics on chip.

[1] Department of Applied Physics, Stanford University, 348 Via Pueblo, Stanford, California 94305, USA. [2] Ginzton Laboratory and Department of Electrical Engineering, Stanford University, Stanford, California 94305, USA. Correspondence and requests for materials should be addressed to S.F. (email: shanhui@stanford.edu).

A Weyl point is a point degeneracy between two bands in a three-dimensional (3D) band structure, with linear dispersion in all three dimensions in its vicinity[1–5]. The simplest Hamiltonian in the wavevector space (k-space) that supports a Weyl point is $\mathcal{H} = v_x k_x \sigma_x + v_y k_y \sigma_y + v_z k_z \sigma_z$. As $\sigma_{x,y,z}$ together with the identity matrix form a complete basis for $2 \times 2$ Hermitian matrices, any perturbation on $\mathcal{H}$ that preserves the translational symmetry can be written as a linear superposition of these four matrices. Thus, any small perturbation in k-space can only shift the Weyl point without destroying the degeneracy and opening a gap[5,6]. Weyl points are 3D topological states: they are monopoles of Berry curvature in the wavevector space[7]. Any closed two-dimensional (2D) surface surrounding the Weyl point has a unit Chern number. This implies the existence of topological surface states, in the form of a Fermi arc connecting two Weyl points of opposite charges for a finite system with its bulk described by $\mathcal{H}$[8–10].

A Weyl point is a 3D object. Therefore, previous works on Weyl points in photonics[11–15], plasmonics[16,17] and acoustics[18,19] have complex 3D geometries, which limits the potential for exploring Weyl point physics in on-chip integrated systems. To explore a Weyl point in a planar 2D geometry, one may use a synthetic dimension[20,21] to simulate the third spatial dimension. The notion of synthetic dimension was previously proposed for superconducting qubits[22], cold atoms[23] and optics[24] based on the idea of increasing local mode connectivity. One can also form the synthetic dimension using the modes of a ring resonator at different frequencies[25–27]. The size of the synthetic dimension, which corresponds to the number of modes in each individual ring, can be rather large without increasing the system complexity.

Here we create a synthetic 3D space by dynamically modulating a 2D array of on-chip ring resonators. Each resonator supports a set of discrete modes equally spaced in resonant frequency. These discrete modes thus form a periodic lattice in the third, synthetic frequency dimension. The two spatial dimensions and one synthetic frequency dimension together form a 3D space. Dynamic modulation of the refractive index leads to effective coupling of modes in the synthetic dimension[25–29]. We show that proper design of the modulation leads to Weyl points in the synthetic space. Our proposed approach is specifically designed for implementation using an existing on-chip integrated photonic platform. Compared with the complex 3D electromagnetic or acoustic structure previously used to demonstrate Weyl point physics[11–19], our approach provides a far more flexible platform to explore a wide range of phase space. For example, by changing the dynamic modulation phases, the same device can be tuned to exhibit line nodes and Weyl points under inversion or/and time-reversal symmetry breaking. This system also provides a novel manifestation of Weyl point physics in terms of a surface state in the synthetic space that exhibits one-way frequency conversion. More generally, in the context of topological photonics[25–39], this work points to the significant richness in using dynamic refractive index modulation to achieve novel topological effects.

## Results

**Model Hamiltonian system and Weyl points.** Our exemplary system consists of a 2D honeycomb array of identical ring resonators as shown in Fig. 1a. In the vicinity of a resonant frequency $\omega_0$, each ring resonator supports a discrete set of resonant modes at frequencies described by $[\beta(\omega_m) - \beta(\omega_0)] \times L = 2m\pi$ ($m = 0, \pm 1, \pm 2 ...$), where $L$ is the circumference of the ring and $\beta$ is the effective wavevector. In the absence of group velocity dispersion, these modes are equally spaced in frequency by the free spectral range $\Omega = 2\pi \frac{v_g}{L}$ ($v_g$ is the group velocity), forming a frequency comb, that is, the $m$-th order sideband

has a frequency $\omega_m = \omega_0 + m\Omega$. We assume static coupling only between modes with the same frequency at the nearest neighbour resonators with an evanescent coupling strength $t_{xy}$. In addition, each resonator is modulated at the frequency $\Omega$, which induces dynamic coupling between modes in the same resonator with frequency separated by $\Omega$, with a coupling strength $t_f$. With appropriate design of the modulation, as detailed in the Supplementary Note 1 and Supplementary Figs 1 and 2, the tight binding Hamiltonian of the system is then

$$
\mathcal{H} = \sum_{i,m} \omega_m a_{i,m}^\dagger a_{i,m} + \sum_{\langle ij \rangle, m} t_{xy} \left( a_{i,m}^\dagger a_{j,m} + a_{j,m}^\dagger a_{i,m} \right)
$$
$$
+ \sum_{i,m} 2 t_f \cos(\Omega t + \phi_i) \left( a_{i,m}^\dagger a_{i,m+1} + a_{i,m+1}^\dagger a_{i,m} \right) \tag{1}
$$

where $i(j)$ labels different resonators in the array and $\phi_i$ is related to the modulation phase on the $i$th resonator. $a$ and $a^\dagger$ denote the standard ladder operators. The second term in $\mathcal{H}$ sums the overall pairs of nearest-neighbour resonators.

Define $c_{i,m} \equiv a_{i,m} e^{-i\omega_m t}$, which represents a transformation to a rotating frame with angular frequency $\omega_m$. Under rotating wave approximation, equation (1) becomes

$$
\mathcal{H} = \sum_{\langle ij \rangle, m} t_{xy} \left( c_{i,m}^\dagger c_{j,m} + c_{j,m}^\dagger c_{i,m} \right)
$$
$$
+ \sum_{i,m} t_f \left( e^{-i\phi_i} c_{i,m+1}^\dagger c_{i,m} + e^{i\phi_i} c_{i,m}^\dagger c_{i,m+1} \right) \tag{2}
$$

We note that the eigenfrequency of the time-independent Hamiltonian in equation (2) corresponds to the quasi-energy of the time-dependent Hamiltonian in equation (1)[28,40]. For the rest of the study, when there is no confusion, we refer to the eigenfrequencies of equation (2) as the 'frequency' of the system.

Without dynamic modulation (that is, $t_f = 0$), the Hamiltonian in equation (2) is block diagonal with respect to index $m$. For each $m$, the Hamiltonian can be transformed to the wavevector space as

$$
\mathcal{H}^{2D}(k_x, k_y) = t_{xy} \left( \cos k_y a + 2\cos \frac{k_y a}{2} \cos \frac{\sqrt{3}}{2} k_x a \right) \sigma_x
$$
$$
+ t_{xy} \left( \sin k_y a - 2 \sin \frac{k_y a}{2} \cos \frac{\sqrt{3}}{2} k_x a \right) \sigma_y \tag{3}
$$

where $(k_x, k_y)$ is the Bloch wavevector in the two spacial dimension space, $a$ is the centre-to-centre separation between nearby resonators and $\sigma_{x,y,z}$ are the three Pauli matrices whose bases are the resonant modes on the two inequivalent $A$ and $B$ sites in the primitive unit cell of the honeycomb lattice. These sites form the $A$ and $B$ sublattices. Equation (3) exhibits Dirac cones at the $K$ and $K'$ points, corresponding to $\mathbf{K}_\pm = \left( \pm \frac{4}{3\sqrt{3}} \frac{\pi}{a}, 0 \right)$ in the reciprocal space, respectively. The effective Hamiltonian at $\mathbf{k} = \mathbf{K}_\pm + \mathbf{q}$ for small $\mathbf{q}$ is

$$
\mathcal{H}^{2D}_{\text{eff}, \mathbf{K}_\pm}(\mathbf{q}) = \frac{3}{2} t_{xy} a \left( \mp q_x \sigma_x + q_y \sigma_y \right)
$$

Next, we consider applying a dynamic refractive index modulation to the rings with the frequency $\Omega$. Such a modulation enables transitions between modes with different $m$. This is captured by the second term in equation (2), which has the form of an effective coupling in the synthetic frequency dimension labelled by $m$. The resulting Hamiltonian therefore represents a tight binding model in 3D, with two spatial dimensions and one synthetic frequency dimension. Although in principle the phase $\phi_i$ can be chosen arbitrarily for each site $i$, to maintain the spatial periodicity of the honeycomb lattice, we only consider the case that $\phi_i = \phi_A$ for sublattice $A$ and $\phi_i = \phi_B$ for sublattice $B$.

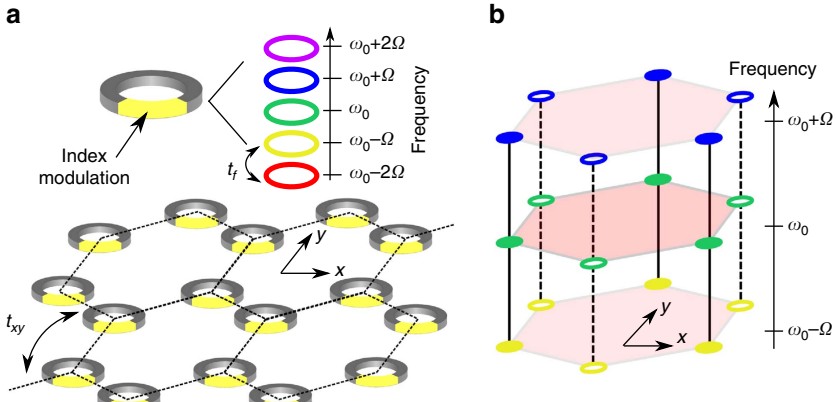

**Figure 1 | A synthetic 3D lattice realized using 2D array of ring resonators.** (**a**) A 2D honeycomb array of ring resonators. Each resonator supports resonant modes evenly-spaced in frequency by $\Omega$. A refractive index modulation with frequency $\Omega$ is applied to each ring resonator. (**b**) A synthetic 3D lattice describing the system in **a**. Filled and hollow circles represent the inequivalent sub-lattice of a honeycomb lattice. Different colours represent resonances of different frequencies. The dashed and solid black vertical links represent coupling along the synthetic frequency dimension generated by refractive index modulation.

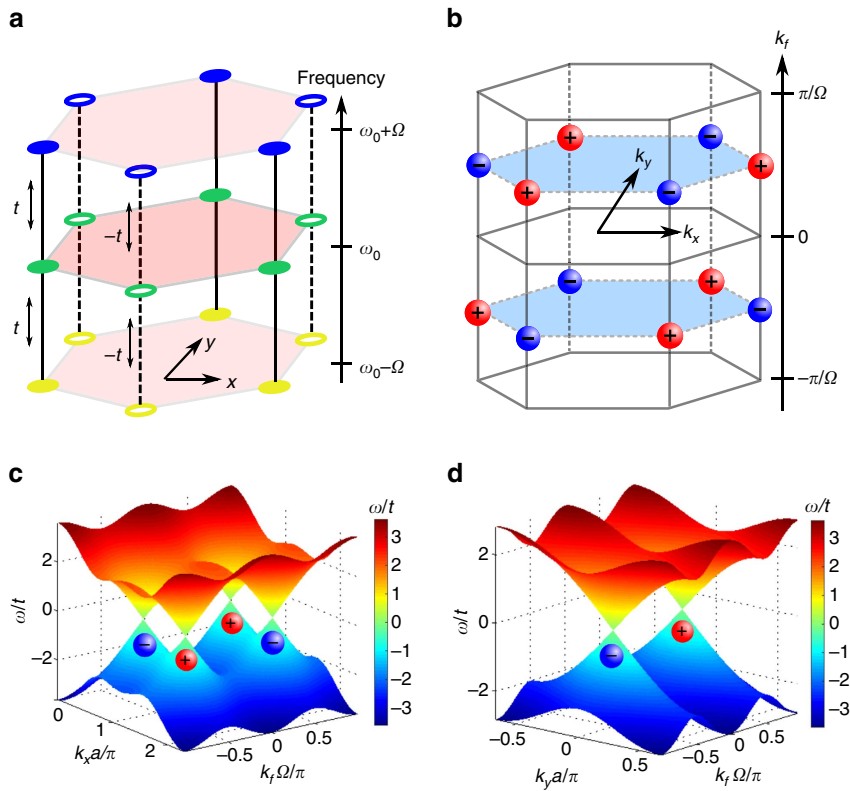

**Figure 2 | Weyl points in an inversion symmetry breaking structure.** (**a**) Modulation phase $\phi_A = 0$, $\phi_B = \pi$ leads to inversion symmetry breaking. (**b**) The Weyl points in the reciprocal space and their charges. The Weyl points are located at $k_y = 0$, $k_x = \pm \frac{4}{3\sqrt{3}}\frac{\pi}{a}$ and $k_f = \pm \frac{\pi}{2\Omega}$. (**c**) The band structure in $k_x - k_f$ plane at $k_y = 0$. The Weyl points at $k_x = \frac{8}{3\sqrt{3}}\frac{\pi}{a}$ are equivalent to those at $k_x = -\frac{4}{3\sqrt{3}}\frac{\pi}{a}$ in **b**. (**d**) The band structure in $k_y - k_f$ plane at $k_x = \frac{4}{3\sqrt{3}}\frac{\pi}{a}$. The charge of each Weyl point is shown underneath.

If $\phi_A = \phi_B$, the coupling constant along the synthetic frequency dimension is spatially homogeneous and we can always choose the gauge with $\phi_i = 0$ for all $i$. The resulting Hamiltonian for the Bloch wavevector $(k_x, k_y, k_f)$ is

$$\mathcal{H}(k_x, k_y, k_f) = 2t_f \cos(k_f\Omega)I + \mathcal{H}^{2D}(k_x, k_y)$$

where $I$ is the $2 \times 2$ identity matrix and $k_f$ is the wavevector in the synthetic dimension. This Hamiltonian exhibits two line nodes at $(k_x, k_y) = \mathbf{K}_\pm$, which are line degeneracies with linear dispersion along $k_x$ and $k_y$. These line nodes are protected by parity time symmetry[5].

Parity time symmetry in the system can be broken by making the modulation phases different for the two sub-lattices, as denoted in Fig. 1b. In the following, we will study two different modulation phase configurations that break either inversion or time-reversal symmetry, thus reducing the line node degeneracy to pairs of Weyl points. Other modulation phase configurations can be chosen to break both inversion and time-reversal symmetry.

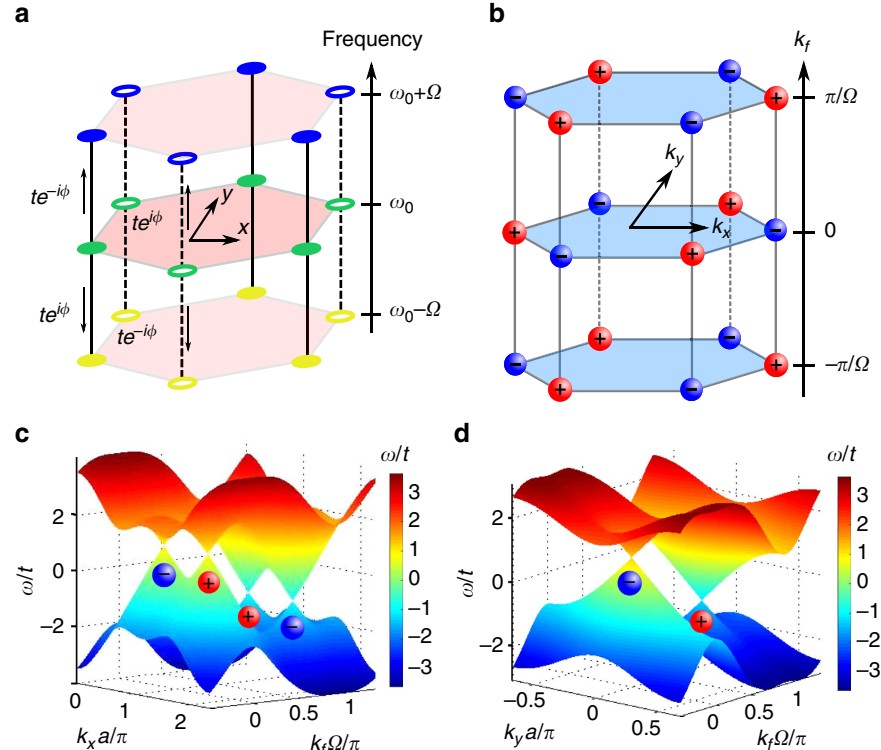

**Figure 3 | Weyl points in a time-reversal symmetry breaking structure.** (**a**) Modulation phase $\phi_A = +\pi/3$ and $\phi_B = -\pi/3$ leads to time-reversal symmetry breaking. (**b**) The Weyl points in the reciprocal space, and their charges. The Weyl points are located at $k_y = 0$, $k_x = \pm \frac{4}{3\sqrt{3}}\frac{\pi}{a}$ and $k_f = 0$ or $\frac{\pi}{\Omega}$. (**c**) The band structure in $k_x - k_f$ plane at $k_y = 0$. The Weyl points at $k_x = \frac{8}{3\sqrt{3}}\frac{\pi}{a}$ are equivalent to those at $k_x = -\frac{4}{3\sqrt{3}}\frac{\pi}{a}$ in **b**. (**d**) The band structure in $k_y - k_f$ plane at $k_x = \frac{4}{3\sqrt{3}}\frac{\pi}{a}$. The charge of each Weyl point is shown underneath.

**Inversion symmetry breaking.** For the system shown in Fig. 2a with a choice of $\phi_A = 0$ and $\phi_B = \pi$, we break inversion symmetry while preserving time-reversal symmetry. Similar $\pi$-flux models where the sign of hopping alternates on the two sublattices have previously been considered in 3D systems[18,41]. Here we provide a different physical implementation. The Hamiltonian in equation (2), for Bloch wavevector $(k_x, k_y, k_f)$, becomes:

$$\mathcal{H}(k_x, k_y, k_f) = 2t_f \cos(k_f \Omega)\sigma_z + \mathcal{H}^{2D}(k_x, k_y)$$

The effective Hamiltonians near $\mathbf{K}_\pm$ and $k_f = \pm\frac{\pi}{2\Omega}$ are:

$$H_{\mathrm{eff},\, \pm\frac{\pi}{2\Omega},\, \mathbf{K}_\pm}(\mathbf{q}) = \left(\mp 2t_f\Omega\right)q_f\sigma_z + \frac{3}{2}t_{xy}a\left(\mp q_x\sigma_x + q_y\sigma_y\right)$$

which have the form $v_x q_x \sigma_x + v_y q_y \sigma_y + v_z q_f \sigma_z$. The band structure along $k_x - k_f$ and $k_y - k_f$ are shown in Fig. 2c,d, confirming the linear dispersion along all three wavevector axes.

The charge of a Weyl point, which is defined as the Berry flux of the lower band in its proximity, is $\mathrm{sgn}(v_x v_y v_z)$[18] and is shown in Fig. 2b. The Weyl points at $(\mathbf{K}_+,\ +\frac{\pi}{2\Omega})$ and $(\mathbf{K}_-,\ -\frac{\pi}{2\Omega})$ have the same charge, as required by time-reversal symmetry[5].

**Time-reversal symmetry breaking.** For the system shown in Fig. 3a with a choice $\phi_A = +\phi$ and $\phi_B = -\phi$, where $\phi \in (0, \pi/2)$ is positive, we break time-reversal symmetry while preserving inversion symmetry. In this case, the coupling along the synthetic frequency axis is directional, as frequency up conversion and down conversion will pick up opposite phases from the refractive index modulation[28].

The Hamiltonian in equation (2), for Bloch wavevector $(k_x, k_y, k_f)$, becomes:

$$\mathcal{H}(k_x, k_y, k_f) = 2t_f \cos\phi \cos(k_f\Omega)I$$
$$+ 2t_f \sin\phi \sin(k_f\Omega)\sigma_z + \mathcal{H}^{2D}(k_x, k_y)$$

Now the Weyl points are at:

$$\omega = 2t_f \cos\phi,\quad \mathbf{k} = (\mathbf{K}_\pm, 0)$$
$$\omega = -2t_f \cos\phi,\quad \mathbf{k} = \left(\mathbf{K}_\pm, \frac{\pi}{\Omega}\right)$$

The effective Hamiltonians near $\mathbf{K}_\pm$ and $k_f = 0$ or $\frac{\pi}{\Omega}$ are:

$$H_{\mathrm{eff},\, 0/\frac{\pi}{\Omega},\, \mathbf{K}_\pm}(\mathbf{q}) = \left(\pm 2t_f\Omega\sin\phi\right)q_f\sigma_z + \frac{3}{2}t_{xy}a\left(\mp q_x\sigma_x + q_y\sigma_y\right)$$

which have linear dispersion along all three wavevector axes, as confirmed by the band structure along $k_x - k_f$ and $k_y - k_f$ plotted in Fig. 3c,d.

In terms of the charge of Weyl points, inversion symmetry requires that Weyl points at $(\mathbf{K}_+, 0)$ and $(\mathbf{K}_-, 0)$ have opposite charges and so as Weyl points at $(\mathbf{K}_+, \frac{\pi}{\Omega})$ and $(\mathbf{K}_-, \frac{\pi}{\Omega})$[5]. This is confirmed by our Weyl point charge calculation shown in Fig. 3b.

**Surface states in analogy with Fermi arcs.** In the previous section, we study Weyl point behaviour in a periodic lattice without truncation. In this section, we demonstrate the surface states in this system when truncated in space. These surface states are one of the key experimental signatures of Weyl point. They are the photonic analogues of the Fermi arc states in electronic systems[42].

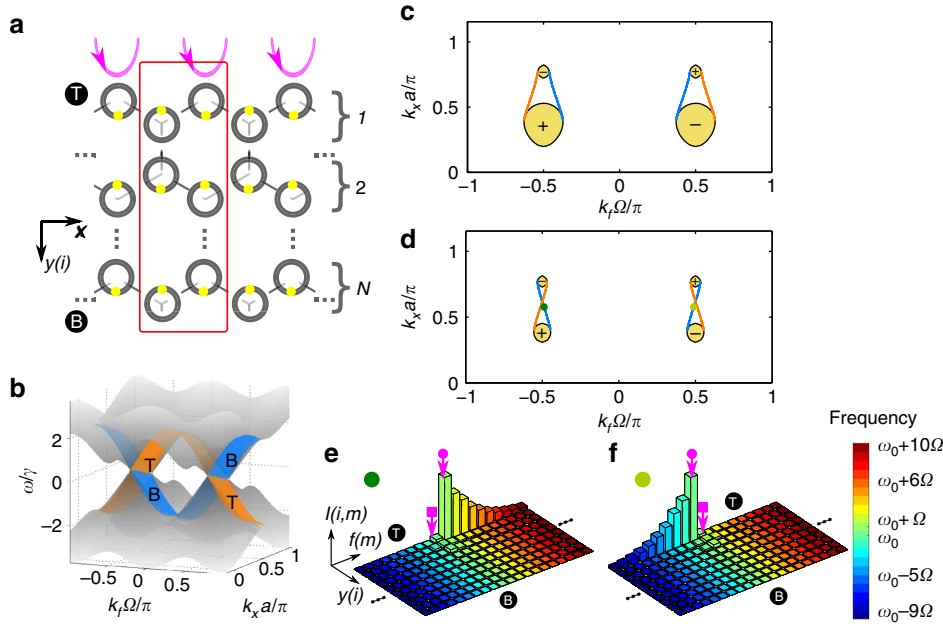

**Figure 4 | Surface states on a stripe with inversion symmetry breaking.** (**a**) The structure consisted of honeycomb array of ring resonator with $N$ rows in $y$ and infinite in $x$. The yellow dots represent the electro-optic modulator. The red box shows a supercell for simulating (**b**–**f**). T and B label the top and bottom edge. Pink arrows indicate input waveguides. (**b**) Band structure showing the bulk states (transparent grey) and the surface states (orange and blue sheets). $N = 40$ rows in $y$ are used in the calculation. The Weyl points at $k_x = \frac{2\pi}{3\sqrt{3}a}$ have $\omega \approx -0.26\gamma$, whereas those at $k_x = \frac{4\pi}{3\sqrt{3}a}$ have $\omega \approx 0.26\gamma$. (**c**) A constant frequency band structure at $\omega = 0.5\gamma$. Bulk states shown in yellow are from the bands above the Weyl points. $+/-$ indicates the charge of Weyl points. Blue and orange lines show the top and bottom surface states. (**d**) A constant frequency band structure at $\omega = 0.07\gamma$. Bulk states shown in yellow are from the band above the Weyl point near $k_x = \frac{2\pi}{3\sqrt{3}a}$ and the band below the Weyl point near $k_x = \frac{4\pi}{3\sqrt{3}a}$. (**e,f**) Static state modal intensity driven by continuous wave input. $N = 5$ rows in $y$ and 60 resonant modes are used. A round trip transmission of 90% for each mode is used to account for losses. Input to the top row of rings consists of two frequencies as marked by pink arrows, with phase delay of $\frac{\pi}{2}$ between the square-ended and circular-ended arrows. The states excited in **e,f** are primarily the ones represented by the dark and light green dot in **d**, respectively, that is, $k_x = \frac{\pi}{\sqrt{3}a}$ and $k_f = \pm \frac{\pi}{2\Omega}$. Only 20 resonances closest to the excitation are shown. The mode intensity of the $A$ and $B$ sublattices in each row are plotted separately along $y$.

The results presented up to now have been obtained using the tight binding model of equation (1), which is a simple model that captures the essence of Weyl point physics in our system. In contrast, in this section, as our objective is to illustrate an experimental signature, we used a more realistic model of a ring resonator under external modulation. In this model[26,34], we consider the dynamics of the circulating amplitudes for each waveguide forming the ring. Both the static coupling between the rings and the modulation are treated using these amplitudes. Below, we refer to this model as the waveguide amplitude model. Details of the calculation can be found in Supplementary Note 2 and Supplementary Fig. 3. Compared with the tight binding model, where there is only nearest-neighbour coupling along the frequency axis, in this waveguide amplitude model there is a long-range coupling beyond the nearest neighbour. Nevertheless, the key characteristics from the tight binding model, in particular the existence and the properties of the Weyl points, are preserved in this waveguide amplitude model, as shown in Supplementary Notes 3 and 4, and Supplementary Figs 4 and 5.

Using the waveguide amplitude model, we calculate the surface states for the structure shown in Fig. 4a. Here we choose the modulation phase $\phi_A = 0$, $\phi_B = \pi$ to break inversion symmetry while preserving time-reversal symmetry. The case where time-reversal symmetry is broken is presented in Supplementary Note 5 and Supplementary Fig. 6. The overall characteristic of the surface state in the two systems are quite similar, as the surface states arise from the Weyl points. The inter-ring coupling $\gamma$ and the modulation strength are chosen such that the nearest-neighbour coupling is isotropic in the spacial and frequency axes, that is, $t_{xy} = t_f$ in the effective tight binding model.

Figure 4b shows the band structure for a stripe infinite along both $x$ and the synthetic frequency dimension; thus, the eigenstates are Bloch states with well-defined wavevector $k_x$ and $k_f$. Two types of states can be easily identified. The bulk states for different $k_y$ are projected inside the four Weyl cones shown in Fig. 2c. In addition, surface states on the top and bottom surface of the stripe (shown in Fig. 4b as transparent blue and orange sheets respectively) are well separated from the bulk states. On a constant frequency cut through the Weyl points, these surface states show up as arcs connecting a pair of Weyl points, in analogy to Fermi arc in electronic systems[42].

Figure 4c,d show constant frequency cuts at two frequencies. On the constant frequency cut, the bulk states form a disk around each of the four Weyl points. The surface states form two pairs of arcs, plotted in blue and orange lines for the top and bottom surface respectively. Unlike the tight binding model in which the pair of Weyl points usually have the same frequency and the surface state arcs connecting them run parallel[42], our calculation using the waveguide amplitude model shows that the pair of Weyl points in our system are different in frequency. Figure 4c is a cut at a frequency above the Weyl points. Its two surface state arcs, although not parallel, do not cross. Figure 4d is a cut at a frequency between the frequencies of the Weyl points. Its two surface state arcs cross, which is not seen in tight binding models.

The surface states on the top and bottom surfaces with the same Bloch wavevector have opposite group velocities along the

frequency dimension. This is evident from the opposite slopes of the orange and blue sheets in Fig. 4b. This implies that a surface state in our system supports one-way frequency conversion. However, in our system there are two pairs of Weyl points and consequently two counter-propagating surface states on the same surface. This is best illustrated in Fig. 4b, where the surface states around the $k_f = \frac{\pi}{2\Omega}$ pair of Weyl points propagate anticlockwise (down-conversion on the top surface and up-conversion on the bottom surface), whereas those around the $k_f = -\frac{\pi}{2\Omega}$ pair of Weyl points propagate clockwise. To observe the one-way frequency conversion of the surface state, the input excitation must be chosen such that it has a $k_x$ that supports a surface state. Furthermore, the input must have maximal coupling to $k_f = \frac{\pi}{2\Omega}$ and minimal coupling to $k_f = -\frac{\pi}{2\Omega}$, or vice versa. This can be achieved by excitation with two frequencies matched to two adjacent resonant modes, with the proper relative phase delay between these two frequency components.

Figure 4e,f present simulation of the one-way frequency conversion of the surface state. Dynamic modulation is introduced through an electro-optic modulation on the ring resonator and a small loss is assumed for each resonator. The input waveguides have a linear delay along $x$ to couple to the $k_x = \frac{\pi}{\sqrt{3}a}$ Bloch state on the top surface of the stripe. The input excitation consists of two nearest resonant frequencies $\omega_0$ and $\omega_0 + \Omega$ with a phase difference of $\pm\frac{\pi}{2}$, which leads to selective coupling to either of $k_f = \pm\frac{\pi}{2\Omega}$. Consequently, Fig. 4e,f correspond to the surface states marked by dark and light green dots in Fig. 4d, respectively. Figure 4e,f demonstrate purely up or down one-way frequency conversion, which provides a measurable experimental signature of the Weyl points and corresponding topological surface states in our proposed system.

The zig-zag edge of a honeycomb lattice also supports a pair of edge states. However, these states are non-chiral and can be easily distinguished from the topologically non-trivial surface states arising from the Weyl points, by examining their propagation direction on opposite edges of a stripe. The zig-zag edge states propagate along the same direction on the opposite edges, whereas the surface states in Weyl point system propagate along different directions. This is verified by the simulation shown in Supplementary Fig. 7.

## Discussion

The main contribution of this manuscript is the realization that three-dimensional topological effects can be demonstrated in a planar structure. It greatly reduces the structural complexity of a 3D photonic lattice to a 2D one. This simplification makes it experimentally viable to demonstrate Weyl point physics using an integrated photonics platform in the infrared and visible wavelength regime, which is a significant step towards using the photonic topological phenomenon in on-chip photonic systems. Furthermore, we demonstrate that frequency space can provide new physics beyond the underlying static system. Our proposed system also allows access to multiple types of topological states in the same device. By changing the dynamic modulation phase, the system can change from exhibiting line nodes to Weyl points under inversion or/and time-reversal symmetry breaking. The general idea of Weyl point in 2D system with a synthetic dimension can also be implemented in other geometrical configurations besides honeycomb lattices[41] and potentially using other types of resonant systems[21].

Our proposed system can be implemented using silicon ring resonator under carrier-depletion modulation[43]. As detailed in Supplementary Note 6 and Supplementary Fig. 8, a $4 \times 3$ array of silicon rings with free spectral range of each ring at 26 GHz as experimentally demonstrated in ref. 44 (which coincide

with the modulation frequency), with an effective coupling constant $t_{xy} = t_f = 8$ GHz and an internal loss rate of 2.7 GHz ($Q = 2 \times 10^5$), suffice to demonstrate the one-way frequency conversion in the Fermi-arc surface state described in the previous section. The entire array can fit into a $3 \times 4$ mm$^2$ chip, which can be further reduced to $1 \times 1$ mm$^2$ by increasing the modulation frequency to 50 GHz and folding of the ring[45], as shown in Supplementary Fig. 9. The required refractive index modulation strength of 0.5 mm$^{-1}$ and optical loss of 1.1 dB mm$^{-1}$ (see Supplementary Table 1), albeit challenging, has been demonstrated in silicon waveguide modulators[46,47]. Other types of resonators and modulation schemes such as LiNO$_3$ microdisks[48] and polymer modulators[49] may also be used. Compared with the silicon ring resonators, these schemes may provide higher quality factors, stronger and intrinsically lossless modulation, and/or higher modulation bandwidth.

**Data availability**. The data that support the findings of this study are available from the corresponding author on request.

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

## Acknowledgements

This work is supported in part by grants from the Air Force Office of Scientific Research, grant numbers FA9550-12-1-0488 and FA9550-12-1-0024. Q.L. acknowledges the support of a Stanford Graduate Fellowship.

## Author contributions

Q.L. conceived the idea of realizing Weyl points in a two dimensional array of ring resonators with synthetic frequency dimension. All authors contributed to the design of the study, discussion of the results and writing of the manuscript. S.F. supervised the project.

## Additional information

**Competing financial interests:** The authors declare no competing financial interests.

**Publisher's note**: 

