## [Peer Review File · Nature Communications]

Reviewers' comments:

Reviewer #1 (Remarks to the Author):

In the manuscript "Photonic Weyl Point in a Two-Dimensional Resonator Lattice with a Synthetic Frequency Dimension," Lin et al. propose a scheme to use coupled photonic ring resonators to realize Weyl points: three-dimensional, stable points with quantized Berry flux. By temporally modulating the rings, the authors theoretically propose to use the mode index in each ring as a synthetic dimension. Since these are discrete and equally spaced in frequency, the system straightforwardly takes on a three-dimensional lattice structure. The modulation also can induce a hopping phase between frequency 'layers' of the honeycomb lattice; these phases can be used to induce a time-reversal or inversion breaking term to the Hamiltonian, thus converting a Dirac line node to a set of Weyl points. The paper is very well written and the topic is certainly contemporary and will be of interest. However I have a number of concerns that preclude me from recommending the paper for publication as is. I outline the concerns below and I would be interested to see the response from the authors.

(1) For typical silicon ring resonators, the modulation frequency via carrier injection is on the scale of 10GHz maximum. Its typical radius would be, say 10 microns, and therefore the free spectral range would be ~ 1000 GHz. Would this not make the coupling to 'neighboring' resonances within a given ring very weak? Can the authors take realistic parameters from a system of their choice (I assume silicon?) and show that before loss takes over they will have significant hopping in the synthetic frequency dimension? I'm very much aware of the work in Refs. [42, 43] but I don't consider a simple reference to these works an answer to my question. The reason is that in [43] the rings are *enormous* (the whole chip?) giving rise to a very small FSR. Creating an array of these would lead to large losses and significant disorder in the resonance frequencies simply from chip height fluctuations. Also, all-optical modulation for a whole array with specially targeted phases is impractical. My question is: given all of these realistic constraints, can the authors convince me that their system is realistically implemented in the lab?

(2) The honeycomb lattice already has edge states even without the third frequency dimension. The authors should comment on how they would be able to conclusively demonstrate in an experiment that they were seeing more than just the honeycomb edge states but the actual Fermi arcs.

(3) I would also like to point out that the authors currently have another recent manuscript on Weyl points: arXiv:1604.00574. This may or may not be under submission currently. While the two proposals are certainly different, in my view each detracts from the impact of the other since they are both proposing the same thing but with different geometries. I simply wish to flag this to the editors and ask the authors to comment.

(4) The notion of synthetic frequency dimensions was pioneered first in ultracold atoms by the group of Lewenstein and then in optics by the group of Buljan. Appropriate in-context citations are warranted:

Boada, O., et al. "Quantum simulation of an extra dimension." *Physical review letters* 108.13 (2012): 133001.

Jukić, Dario, and Hrvoje Buljan. "Four-dimensional photonic lattices and discrete tesseract solitons." *Physical Review A* 87.1 (2013): 013814.

Reviewer #2 (Remarks to the Author):

In this manuscript, the authors propose how to realize and explore Weyl points in integrated photonics, with the aim of exploiting the resulting topological surface states for one-way frequency conversion. In their proposal, a two-dimensional honeycomb array of ring resonators is combined with a "synthetic" frequency dimension to form an effectively three-dimensional lattice. By carefully designing the couplings between rings and modes, the authors show that this system can host Weyl points with either inversion or time-reversal symmetry, and study how the corresponding surface state arcs could be excited experimentally on chip.

Overall, this is a reasonable proposal that combines two hot topics of current research, namely Weyl points and synthetic dimensions. It is not surprising that these two directions can be successfully combined, and indeed, this seems to follow as a very natural next step given previous work in the field, as cited also in the manuscript. With this in mind, the innovation of this proposal is not particularly striking but I believe it is sufficiently simple and versatile that it can be useful to the field. In general, the main results are novel and well presented, although the text contains many small language typos. The work is also backed up by an extensive Supplementary Information for the interested reader. In my opinion, this work could be suitable for publication in Nature Communications after appropriate revisions.

I have a few hesitations, especially concerning the discussion of topological surface states for one-way frequency conversion. Firstly, the simulations shown in Figure 4 to support this point involve a lattice which is infinite along the x direction? However, this is not experimentally realistic; could the authors address the issue of finite-size effects also along this direction? How large a system is required along both x and y to exploit these states?

Secondly, in some of the authors' previous work (Ref. [24] in this manuscript), they also proposed how topological edge states in a synthetic dimension could be used for frequency conversion. How does this compare with the current scheme for practical applications? Indeed the scheme in this manuscript appears to have comparative disadvantages such as requiring larger system sizes? Continuing the issue of finite-size effects from above, Ref. [24] also limits the number of modes considered to 11 side bands, whereas here 60 resonator modes were used. Which is more physically realistic? What happens to the surface states studied here if the light hits the effective "boundary" in frequency space?

As a final suggestion, I would also recommend that the authors add the following two references

[1] W.-Y He et al., Phys. Rev. A 94, 013606 (2016) on Weyl points in honeycomb lattices.

[2] T. Ozawa et al., arXiv:1607.00140 on synthetic dimensions in integrated photonics.

Reviewer #3 (Remarks to the Author):

In the manuscript "Photonic Weyl Point in a Two-Dimensional Resonator Lattice 2 with a Synthetic Frequency Dimension" the authors suggest a novel approach for implementing a three-dimensional topological photonic state using frequency as an additional "synthetic" dimension. Topological photonics indeed is of great interest to scientific community at present, and implementation of topological systems for electromagnetic and optical modes has been done using various platforms. One of the approaches relies on modulation in time, sometimes referred to as "Floquet topological insulators". In the present work, however, the authors give a completely new twist to this approach and utilize it not only to engineer a topological state, but they in fact predict the existence of the new one, which is typically found in systems with higher dimensionality. This idea is very promising both from scientific and applied points of view as, first, it enables testing fundamental concepts of topological photonics in systems with lower dimensionality and, second, it also envisions absolutely novel approaches to control light by confining it not only along edges in space, but also to the abstract edge in the frequency domain – the synthetic dimension. It is very

important that in contrast to previous approaches to 3D topology, which used overly complicated systems hardly scalable to visible domain, the current work suggests that such states, in particular Weyl points, can be realized on-chip using a conventional silicon photonics platform. I am sure that the present work will be not only highly cited, but also will result in a brand new direction in the field of topological physics, well beyond photonics. I therefore would like to highly recommend the present work in Nature Communications. I would also like to point out to a few minor issues which would preferably be fixed before the manuscript go into production:

1. Line 161: "in the two system are" should be "in the two systems are"
2. Fig. 4b is not clear, please change viewing angle to better reveal the edge states
3. The relevant reference could be a good addition for the reader interested in topology in time-modulated systems: Nature Communications 7, 11744 (2016).

Summary of Changes:

Main Text

1. We added the following sentences to the end of second paragraph in introduction to point out earlier works on synthetic dimension, in response to reviewer #1's request.

The notion of synthetic dimension was previously proposed for superconducting qubits [27], cold atoms [28] and optics [29] based on the idea of increasing local mode connectivity. Here we form the synthetic dimension using the modes of a ring resonator at different frequencies. The size of the synthetic dimension, which corresponds to the number of modes for each individual ring in our system, can be rather large without increasing the system complexity.

2. We significantly modified the last paragraph to include an example experimental system design based on silicon ring resonator with carrier-depletion modulation, and point out other possible platforms for realizing our proposal.

Our proposed system can be implemented using silicon microring under carrier-depletion modulation [43]. As detailed in Supplementary Sec. S6, a 4×3 array of silicon rings with free spectral range of each ring at 26 GHz as experimentally demonstrated in [44] (which coincide with the modulation frequency), an effective coupling constant $t_{xy} = t_f = 8$ GHz and an internal loss rate of 2.7 GHz ($Q=2 \times 10^5$) suffice to demonstrate the one-way frequency conversion in the Fermi-arc surface state described in the previous section. The required refractive index modulation strength of 0.5 mm^{-1} and optical loss of 1.1 dB mm^{-1} , albeit challenging, has been demonstrated in Si waveguide modulators [43, 45, 46]. The entire array can fit into a $3 \times 4 \text{ mm}^2$ chip, which can be further reduced to $1 \times 1 \text{ mm}^2$ by increasing the modulation frequency to 50 GHz and folding of the ring. Other types of resonators and modulation schemes such as LiNO_3 microdisks [47] and polymer modulators [48] may also be used. Compared with the silicon ring resonators under electrical modulation, these schemes may provide higher quality factor, stronger and intrinsically lossless modulation, and/or higher modulation bandwidth. The general idea of Weyl point in 2D system with synthetic dimension can also be implemented in other geometrical configurations besides honeycomb lattice [41], and potentially using other types of resonant systems [21].

3. We included new references [3], [24], [27-29], [39], [43], [45-48].

4. In line 165, we corrected a grammatical error, changing “in the two system” to “in the two systems”.

5. We have change the viewing angle and transparency of Fig. 4(b) to better present the surface states, as suggested by Reviewer 2.

6. We have added titles for all figures.

Supplementary

1. We added Section S6. Design of an experimental system. This section provides a detailed analysis of an experimental system based on silicon ring modulator. It also includes simulation result using realistic parameters on a finite array of 12 resonators, and discussion of the effect of frequency boundary on the propagation of edge states.

2. We added Section S7. Comparison with trivial edge state in honeycomb lattice zig-zag edge. This section provides a comparison between topologically trivial zig-zag edge state and Fermi-arc edge state

in Weyl point system using simulation.

3. We have added titles for all figures and tables.

Response to Reviewer Comments:

Reviewer #1:

In the manuscript “Photonic Weyl Point in a Two-Dimensional Resonator Lattice with a Synthetic Frequency Dimension,” Lin et al. propose a scheme to use coupled photonic ring resonators to realize Weyl points: three-dimensional, stable points with quantized Berry flux. By temporally modulating the rings, the authors theoretically propose to use the mode index in each ring as a synthetic dimension. Since these are discrete and equally spaced in frequency, the system straightforwardly takes on a three-dimensional lattice structure. The modulation also can induce a hopping phase between frequency ‘layers’ of the honeycomb lattice; these phases can be used to induce a time-reversal or inversion breaking term to the Hamiltonian, thus converting a Dirac line node to a set of Weyl points. The paper is very well written and the topic is certainly contemporary and will be of interest.

However I have a number of concerns that preclude me from recommending the paper for publication as is. I outline the concerns below and I would be interested to see the response from the authors.

We thank the reviewer for pointing out the merits of our manuscript. We will address the concerns below.

(1) For typical silicon ring resonators, the modulation frequency via carrier injection is on the scale of 10GHz maximum. Its typical radius would be, say 10 microns, and therefore the free spectral range would be ~ 1000 GHz. Would this not make the coupling to ‘neighboring’ resonances within a given ring very weak? Can the authors take realistic parameters from a system of their choice (I assume silicon?) and show that before loss takes over they will have significant hopping in the synthetic frequency dimension? I’m very much aware of the work in Refs. [42, 43] but I don’t consider a simple reference to these works an answer to my question. The reason is that in [43] the rings are *enormous* (the whole chip?) giving rise to a very small FSR. Creating an array of these would lead to large losses and significant disorder in the resonance frequencies simply from chip height fluctuations. Also, all-optical modulation for a whole array with specially targeted phases is impractical. My question is: given all of these realistic constraints, can the authors convince me that their system is realistically implemented in the lab?

The reviewer is right in pointing out that for electrical modulation in silicon waveguide using carrier-injection or depletion, the maximum modulation frequency, which have to coincide with the free spectral range of the resonator, can only be a few tens of GHz, which means the length of the resonator needs to be in the order of a few millimeters, or the radius of the ring on the order of a few hundred microns. Thus a relatively large resonator is needed. However, as we show in the last paragraph of the main text and in Supplementary Section S6, only a small array of 4 by 3 resonators is needed for demonstrating the signature Fermi arc edge state in our system. With some careful design the system may be able to fit into a 1mm by 1mm chip. Given the relatively large size and small numbers of resonator, thermal tuning may be possible for precisely aligning the resonance.

As for the effect of loss, we have also shown that with a coupling strength of 0.5/mm between adjacent modes in frequency space, and a loss of 1.1dB/mm (including passive waveguide loss of 0.1dB/mm and carrier absorption loss of 1dB/mm), we can achieve an effective coupling constant of 8GHz and a loss rate of 2.7GHz in a system with a free spectra range (FSR) of 26GHz. As a result, the 4 by 3 resonator system is sufficient for the demonstration of one-way frequency conversion over 7 resonant modes before the intensity falls below 1/10 of peak intensity. Both the coupling strength and loss parameter has been demonstrated in carrier-depletion modulation in Si waveguide. A detailed description of this design has been added in Supplementary Section S6.

The loss rate as quoted in the paragraph above corresponds to a quality factor of $Q=200,000$. Such a quality factor may be on the higher end for Si ring modulators due to the intrinsically lossy refractive index modulation mechanism in silicon. However, there are other types of resonators that may also be promising for demonstration of our proposal. For example, LiNbO₃ microdisks has intrinsic Q over 1,000,000 and intrinsically lossless electro-optical modulation. Such microdisk systems may therefore be a promising alternative platform for as a proof-of-principle experiment. Another example of a promising platform would be polymer ring resonators, owing to their large modulation bandwidth and strong coupling strength.

(2) The honeycomb lattice already has edge states even without the third frequency dimension. The authors should comment on how they would be able to conclusively demonstrate in an experiment that they were seeing more than just the honeycomb edge states but the actual Fermi arcs.

Honeycomb lattice with zig-zag edge does have edge states, but these edge states are non-chiral, and will have the same direction of frequency conversion on opposite edges of a stripe. In contrast, the Fermi arc edge states are chiral, and will propagate in opposite directions on different edges of the stripe. The comparison of these two kinds of edge states can easily be observed in our system by setting different modulation phases for the two sub-lattices. With $\Phi_A=\Phi_B$, i.e. with identical modulation phases for the two sub-lattices, the system does not have Weyl points, and has only non-chiral edge states originated from that of the zig-zag edge. When exciting two frequencies with phase differences of $\pi/2$, the top and bottom edges both exhibit frequency down conversion. On the other hand, with $\Phi_A-\Phi_B=\pi$, the system has Weyl points due to broken inversion symmetry, and consequently the stripe supports chiral Fermi arc edge states. When exciting with the same input, the top exhibits frequency down-conversion while the bottom edge exhibits frequency up-conversion. In addition, Fermi arc states have better edge confinement and longer propagation distance in the presence of loss as compared to the zig-zag edge states.

We added simulation result comparing these two kinds of edge states in the Supplementary Section S7.

(3) I would also like to point out that the authors currently have another recent manuscript on Weyl points: arXiv:1604.00574. This may or may not be under submission currently. While the two proposals are certainly different, in my view each detracts from the impact of the other since they are both proposing the same thing but with different geometries. I simply wish to flag this to the editors and ask the authors to comment.

The referred paper titled “Hyperbolic Weyl Point in Reciprocal Chiral Metamaterials” has recently been published in Physical Review Letters. We have updated the reference to the published version of this paper. This paper concerns the existence and properties of Weyl points in 3-dimensional reciprocal chiral metamaterials.

In contrast, the manuscript under consideration here focus on achieving Weyl point in a 2-dimension ring resonator array system with a synthetic frequency dimension. While both paper concern Weyl points, they are based on very different physical model and different experimental platforms. Most importantly, the present manuscript points out the possibility of achieving Weyl point physics in two-dimensional on-chip systems, which has never been noted before in any previous works on Weyl point physics including our own previous works.

(4) The notion of synthetic frequency dimensions was pioneered first in ultracold atoms by the group of

Lewenstein and then in optics by the group of Buljan. Appropriate in-context citations are warranted: Boada, O., et al. "Quantum simulation of an extra dimension." *Physical review letters* 108.13 (2012): 133001.

Jukić, Dario, and Hrvoje Buljan. "Four-dimensional photonic lattices and discrete tesseract solitons." *Physical Review A* 87.1 (2013): 013814.

We thank the reviewer for pointing out these references. We have added both references to earlier works on synthetic dimension at the end of second paragraph of the main text.

Reviewer #2:

In this manuscript, the authors propose how to realize and explore Weyl points in integrated photonics, with the aim of exploiting the resulting topological surface states for one-way frequency conversion. In their proposal, a two-dimensional honeycomb array of ring resonators is combined with a "synthetic" frequency dimension to form an effectively three-dimensional lattice. By carefully designing the couplings between rings and modes, the authors show that this system can host Weyl points with either inversion or time-reversal symmetry, and study how the corresponding surface state arcs could be excited experimentally on chip.

Overall, this is a reasonable proposal that combines two hot topics of current research, namely Weyl points and synthetic dimensions. It is not surprising that these two directions can be successfully combined, and indeed, this seems to follow as a very natural next step given previous work in the field, as cited also in the manuscript. With this in mind, the innovation of this proposal is not particularly striking but I believe it is sufficiently simple and versatile that it can be useful to the field. In general, the main results are novel and well presented, although the text contains many small language typos. The work is also backed up by an extensive Supplementary Information for the interested reader. In my opinion, this work could be suitable for publication in *Nature Communications* after appropriate revisions.

We thank the reviewer for the positive feedbacks.

I have a few hesitations, especially concerning the discussion of topological surface states for one-way frequency conversion. Firstly, the simulations shown in Figure 4 to support this point involve a lattice which is infinite along the x direction? However, this is not experimentally realistic; could the authors address the issue of finite-size effects also along this direction? How large a system is required along both x and y to exploit these states?

We have added simulation results with a finite-size system (a 4 by 3 array of ring resonator) in the Supplementary Section 7. The one-way frequency conversion can still be clearly seen in this finite-size system.

Secondly, in some of the authors' previous work (Ref. [24] in this manuscript), they also proposed how topological edge states in a synthetic dimension could be used for frequency conversion. How does this compare with the current scheme for practical applications? Indeed the scheme in this manuscript appears to have comparative disadvantages such as requiring larger system sizes? Continuing the issue of finite-size effects from above, Ref. [24] also limits the number of modes considered to 11 side bands, whereas here 60 resonator modes were used. Which is more physically realistic? What happens to the surface states studied here if the light hits the effective "boundary" in frequency space?

We agree that in terms of practicality for one-way frequency conversion, Ref. [24] is a simpler system.

However, the current system allows changing of the direction of frequency conversion on the same edge by switching the relative phases of the two input frequencies.

As for the number of sidebands involved, in real system this is likely to be determined by the loss of the ring resonator. In Supplementary Section 6 we provide a realistic design based on Si ring resonator, and show that the intensity drops below 10% after propagating over 7 sidebands. Therefore, we could have well used only 20 sidebands in our simulation. On the other hand, given the free-spectral-range of a few tens of GHz, even 60 sidebands would only mean about 3nm from the central frequency at 1550nm, within which the modes can well be considered evenly spaced in frequency. Thus simulating with 60 sidebands is accurate, and does reduce the artifact due to hard frequency cutoff.

In our simulation, the loss is large enough that the fields do not propagate to the boundary along the frequency axis. On the other hand, if the loss is low enough and there is sharp boundary along the frequency axis, then upon reaching the frequency boundary there will always be scattering into bulk states, since in our system the bulk is not gapped. In practice, the scattering into bulk may be small due to the small bulk density of state near Weyl point. In addition, the system with time reversal symmetry supports two counter-propagating Fermi arc states on the same edge, so there will also be back-scattering. A portion of the energy will also take a sharp bend the start propagating along the spacial axis. The second last paragraph in Supplementary Section S6 is added to discuss what happens at the frequency boundary.

As a final suggestion, I would also recommend that the authors add the following two references

[1] W.-Y He et al., Phys. Rev. A 94, 013606 (2016) on Weyl points in honeycomb lattices.

[2] T. Ozawa et al., arXiv:1607.00140 on synthetic dimensions in integrated photonics.

We thank the author for bringing up these references, and have added them to our paper.

Reviewer #3:

In the manuscript “Photonic Weyl Point in a Two-Dimensional Resonator Lattice 2 with a Synthetic Frequency Dimension” the authors suggest a novel approach for implementing a three-dimensional topological photonic state using frequency as an additional “synthetic” dimension. Topological photonics indeed is of great interest to scientific community at present, and implementation of topological systems for electromagnetic and optical modes has been done using various platforms. One of the approaches relies on modulation in time, sometimes referred to as “Floquet topological insulators”. In the present work, however, the authors give a completely new twist to this approach and utilize it not only to engineer a topological state, but they in fact predict the existence of the new one, which is typically found in systems with higher dimensionality. This idea is very promising both from scientific and applied points of view as, first, it enables testing fundamental concepts of topological photonics in systems with lower dimensionality and, second, it also envisions absolutely novel approaches to control light by confining it not only along edges in space, but also to the abstract edge in the frequency domain – the synthetic dimension. It is very important that in contrast to previous approaches to 3D topology, which used overly complicated systems hardly scalable to visible domain, the current work suggests that such states, in particular Weyl points, can be realized on-chip using a conventional silicon photonics platform. I am sure that the present work will be not only highly cited, but also will result in a brand new direction in the field of topological physics, well beyond photonics. I therefore would like to highly recommend the present work in Nature Communications.

We thank the reviewer for the recommendation.

I would also like to point out to a few minor issues which would preferably be fixed before the manuscript go into production:

1. Line 161: “in the two system are” should be “in the two systems are”
2. Fig. 4b is not clear, please change viewing angle to better reveal the edge states
3. The relevant reference could be a good addition for the reader interested in topology in time-modulated systems: Nature Communications 7, 11744 (2016).

We have incorporated these suggestions in the modified manuscript.

REVIEWERS' COMMENTS:

Reviewer #1 (Remarks to the Author):

I find that the authors have addressed my concerns and I recommend publication.

Reviewer #2 (Remarks to the Author):

I would like to thank the authors for responding to my previous suggestions, and making these improvements to the manuscript. I am happy with their changes and the addition of new Sections in the Supplemental Material, in particular dealing with realistic experimental design. I recommend this manuscript for publication in Nature Communications.

Response to Reviewers' Comments:

Reviewer #1 (Remarks to the Author):

I find that the authors have addressed my concerns and I recommend publication.

We thank the reviewer for the recommendation.

Reviewer #2 (Remarks to the Author):

I would like to thank the authors for responding to my previous suggestions, and making these improvements to the manuscript. I am happy with their changes and the addition of new Sections in the Supplemental Material, in particular dealing with realistic experimental design. I recommend this manuscript for publication in Nature Communications.

We thank the reviewer for the recommendation.